# Diagnosis of Bloodstream Infections: An Evolution of Technologies towards Accurate and Rapid Identification and Antibiotic Susceptibility Testing

**DOI:** 10.3390/antibiotics11040511

**Published:** 2022-04-12

**Authors:** Kristel C. Tjandra, Nikhil Ram-Mohan, Ryuichiro Abe, Marjan M. Hashemi, Jyong-Huei Lee, Siew Mei Chin, Manuel A. Roshardt, Joseph C. Liao, Pak Kin Wong, Samuel Yang

**Affiliations:** 1Department of Emergency Medicine, Stanford University School of Medicine, Palo Alto, CA 94305, USA; ktjandra@stanford.edu (K.C.T.); nikhilrm@stanford.edu (N.R.-M.); ryuabe@stanford.edu (R.A.); marjan.mhashemi@gmail.com (M.M.H.); 2Department of Biomedical Engineering, The Pennsylvania State University, University Park, PA 16802, USA; gary19921119@gmail.com (J.-H.L.); spc6112@psu.edu (S.M.C.); mar6320@psu.edu (M.A.R.); pxw28@psu.edu (P.K.W.); 3Department of Urology, Stanford University School of Medicine, Stanford, CA 94305, USA; jliao@stanford.edu; 4Veterans Affairs Palo Alto Health Care System, Palo Alto, CA 94304, USA; 5Department of Mechanical Engineering, The Pennsylvania State University, University Park, PA 16802, USA; 6Department of Surgery, The Pennsylvania State University, Hershey, PA 17033, USA

**Keywords:** infectious diseases, sepsis, pathogen diagnosis, antibiotic susceptibility, emerging technologies, sample preparation, multidrug-resistant pathogens

## Abstract

Bloodstream infections (BSI) are a leading cause of death worldwide. The lack of timely and reliable diagnostic practices is an ongoing issue for managing BSI. The current gold standard blood culture practice for pathogen identification and antibiotic susceptibility testing is time-consuming. Delayed diagnosis warrants the use of empirical antibiotics, which could lead to poor patient outcomes, and risks the development of antibiotic resistance. Hence, novel techniques that could offer accurate and timely diagnosis and susceptibility testing are urgently needed. This review focuses on BSI and highlights both the progress and shortcomings of its current diagnosis. We surveyed clinical workflows that employ recently approved technologies and showed that, while offering improved sensitivity and selectivity, these techniques are still unable to deliver a timely result. We then discuss a number of emerging technologies that have the potential to shorten the overall turnaround time of BSI diagnosis through direct testing from whole blood—while maintaining, if not improving—the current assay’s sensitivity and pathogen coverage. We concluded by providing our assessment of potential future directions for accelerating BSI pathogen identification and the antibiotic susceptibility test. While engineering solutions have enabled faster assay turnaround, further progress is still needed to supplant blood culture practice and guide appropriate antibiotic administration for BSI patients.

## 1. Challenges in Bloodstream Infection Diagnosis

The presence of viable microorganisms bacteria in the blood, i.e., bacteremia, when not controlled properly can lead to the development of bloodstream infection (BSI) and sepsis, a syndromic inflammatory response that contributes to a leading cause of death worldwide [1]. The survival rate of patients with sepsis drops by almost 8% per hour of delayed treatment [2]. The ability to rapidly identify invading pathogens and initiate appropriate treatment is critical yet challenging, primarily due to the typically low pathogen load (1–100 CFU/mL of blood), breadth of pathogen coverage, and the complex blood matrix [3,4]. Meanwhile, the administration of effective antibiotic treatment greatly depends on knowing the pathogens’ identity and antibiotic susceptibility profile. This ongoing issue leads to poor clinical outcomes for BSI patients [5]. The lack of accurate and rapid techniques for the timely elucidation of causative pathogens necessitates the use of broad-spectrum antibiotic agents. The 2021 report by the CDC showed that 28% of antibiotic prescriptions in the United States were unnecessary [6]. This continued misuse of antibiotics can worsen the clinical outcome and will exacerbate the emergence of antibiotic-resistant microorganisms worldwide [7]. 

This review highlights both the progress and shortcomings of current and emerging diagnostic tools in BSI. Key technological advancements that could pave the way for supplanting conventional blood culture practices will be discussed. Our goal is to provide an assessment of the potential future development of pathogen identification (ID) and antibiotic susceptibility tests (ASTs) to address this urgent clinical need.

## 2. Overview of Current Blood Culture Diagnosis Workflow

Clinical presentations of BSI are often vague and varied, making timely clinical diagnoses challenging. However, when BSI is suspected, sampling for blood culture as the laboratory diagnostic standard is usually obtained, followed by immediate, empirical broad-spectrum antibiotic treatment. Once a positive culture is detected, subsequent species ID and AST are performed.

Most of the current ID workflows still rely on a positive blood culture sample. Generally, when a blood culture flags as positive, a further overnight incubation on agar plates (subculture) is performed to obtain pure isolated colonies. Colonies are subjected to Gram stain, followed by a series of biochemical and molecular tests, and/or the matrix-assisted laser desorption ionization time of flight mass spectrometry (MALDI-TOF MS) to confirm the bacterial species. A quantitative measure of antibiotic susceptibility using methods like broth microdilution or ETEST gradient strip (bioMérieux) usually takes place alongside or after ID to give minimum inhibitory concentration (MIC) information of antibiotics against the isolated pathogen. Depending on the growth rate of the microorganisms, it could take anywhere between 24 h and 5 days for blood culture to flag positively. Fully automated systems, such as the (bioMérieux) VITEK 2, are also commonly used for analyzing positive blood cultures, delivering faster AST turnarounds of approximately 9 to 18 h, depending on the organism [8,9]. Overall ID and AST performed after blood culture can subsequently take days or even weeks to complete (Figure 1) [10]. 

Since some broad-spectrum antibiotics are administered as frequently as every 6 to 8 h, clinicians could re-evaluate their choice of antibiotics during this period, based on patients’ conditions. The de-escalation of antibiotics to the causative pathogen should be done as soon as possible, but antibiotic adjustment is less likely when the time to AST result is delayed [11]. In some instances, knowing the causative species could prompt an escalation in antibiotic use, such as implementing double coverage for *Pseudomonas* species or adding colistin for *Acinetobacter* species. Ideally, antibiotic adjustments should be made as early as the second dose of antibiotic administration to reduce any unnecessary use of broad-spectrum antibiotics. In particular, AST results that could be obtained within the same clinical shift are critical so that clinicians could decide on appropriate patient management [12]. Hence, the future development of ID/AST methods should take into account these clinical timepoints, while aiming to deliver results in less than 6 h.

## 3. Challenges of BSI Diagnosis and Clinical Significance for a Novel Technique

Meeting the diagnostic needs of BSI within the acute care timescale would require direct testing from whole blood with both ID and AST capabilities. Ideally, next-generation BSI diagnostics should possess four key elements:(1)Sample preparation directly from whole blood that requires minimal handling and can process sufficient blood volume to ensure the capture of pathogens in low abundance;(2)Workflow that effectively separates, enriches, and concentrates pathogens or target analytes from background interferences;(3)Sensitive, quantitative, and accurate detection and species-level identification that differentiates pathogens from contaminants or commensals;(4)Timely and universal antibiotic susceptibility profiling independent of resistant mechanisms with MIC, reporting matches with critical clinical decision timepoints.

Previous reviews have captured a broad range of pathogen ID and AST technologies that are either in development or clinically available [13,14,15]. Methods that utilize techniques such as atomic force microscopy, surface plasmon resonance, electrochemical impedance, asynchronous magnetic bead, single-cell morphological imaging, and bacteriophage-based methods have also previously been described [14,16,17,18,19,20,21]. While ID and AST do not necessarily have to be performed on a single platform, fully integrated platforms will simplify clinical workflows by requiring less sample handling. In Table 1, we summarized some of these technologies that could perform either ID or AST. These methods either target bacterial cells or their genetic contents and can be implemented either directly from whole blood or from positive blood culture samples. Those that are still in the early development stage (proof-of-concept) are not included.

While current approaches are still short of meeting all the key elements of an ideal diagnostic, recent innovations exploring novel avenues could potentially fill the technological gaps lacking in previous developments. In this section, we discuss the challenges and considerations of fulfilling these four elements, and highlight some emerging technologies that have the potential to address them. 

### 3.1. Sample Preparation and Assay Workflow for Capturing, Separating, and Enriching Low-Abundance Pathogens 

The first two elements of a next-generation BSI diagnosis should effectively and efficiently capture and separate pathogens from whole blood. Sample preparation is one of the most challenging aspects of the diagnostic workflow of infectious diseases [22,23]. Yet, the overall workflow of a BSI diagnosis heavily depends on this process. 

Every milliliter of infected blood carries about 4 to 6 × 10^9^ red blood cells, up to 1.6 × 10^7^ white blood cells, and 1.3 to 4 × 10^8^ platelets, with only 1 to 100 bacteria cells [24,25]. Traditionally, this low abundance of pathogens is overcome by inoculating blood in nutrient-rich broth prior to diagnosis. However, this is a process that could take days and cause severe delays in the overall assay. With the development of advanced amplification and detection techniques, bypassing the lengthy blood culture step by isolating and concentrating pathogens directly from patients’ blood samples will allow dramatic time reduction in the workflow, and may enable a new generation of culture-free BSI diagnostic approaches. 

Table 2 summarizes microfluidic methods for isolating bloodborne pathogens and highlights their performance in selected examples. These pathogen isolation techniques can be broadly classified into chemical and physical methods. Chemical approaches, such as affinity capture and erythrocyte lysis, rely on the biochemical properties of bacteria and blood cells for the positive and negative selection of cells in the samples. Physical approaches, such as acoustics, electrokinetics, hydrodynamics, magnetics, and filtering, isolate bacteria from blood cells based on their differences in size, density, and other physical properties. This section highlights several promising pathogen isolation methods (Figure 2). We pay specific attention to microfluidic sample preparation methods, which are amenable to automation and system integration. 

#### 3.1.1. Affinity Capture

Affinity capture techniques with chemically modified particles and surfaces are popular strategies for selective cell isolation. Biochemical binding with microbeads provides a mechanism for converting chemical features to physical properties for capturing pathogens. For example, surface-modified microbeads can bind and isolate bacteria by magnetic force, gravitation/sedimentation, and filtering [46,47]. Immunoassays with antibody conjugation on microbeads can be applied for isolating specific pathogens and have been implemented in microfluidic formats with 70–80% recovery efficiency at 10–20 mL/h [26,27,48,49]. Furthermore, antibiotics that interact with the bacterial cell wall can be applied for capturing bacteria. For instance, vancomycin and daptomycin interact and inhibit cell wall synthesis for Gram-positive bacteria [50,51]. They were modified as affinity probes on the surfaces of magnetic microbeads for isolating Gram-positive bacteria [52,53]. In addition, glycosaminoglycans, such as heparin or heparan sulfate, are widely distributed in all human tissues, and some bacteria bind specifically to them [54]. Capturing *S. aureus* from blood with over 65% efficiency has been demonstrated using surface-heparinized polyethylene microbeads [55]. These strategies are promising when there is a specific group of target pathogens. Nevertheless, finding a ligand that binds strongly and selectively to all pathogens, but not other blood components, can be challenging. In clinical scenarios where the bacterial species is unknown, multiple probes may be required to cover the large range of pathogens associated with BSI, and to avoid a false negative.

Efforts have also been devoted to the broad-spectrum capture of pathogens. In the human immune system, mannose-binding lectin (MBL) binds the exterior of pathogens and pathogen-associated molecular patterns that activate the innate immune response [41]. MBL and engineered MBL are promising approaches for the sample preparation of BSI diagnostics. Owing to their broad binding ability to pathogens, MBLs are commonly applied to bacteria capture and blood cleansing [28,33,56]. Additionally, zinc-coordinated bis(dipicolylamine) (Zn-DPA) that binds to both Gram-positive and Gram-negative bacteria is another promising candidate of broad-spectrum capture probes [30]. By using a multistage microfluidic device, Lee et al. demonstrated the removal of >95% bacteria at 60 mL/h for blood cleansing [30]. Challenges of affinity capture include false negatives associated with probe specificity and viscous drag caused by blood components. These challenges can limit the overall throughput and capture efficiency of affinity capture approaches. 

#### 3.1.2. Erythrocyte Depletion

Erythrocyte depletion is an important strategy for pathogen isolation. Negative selection reduces the complexity of the sample and enhances the efficiency of downstream enrichment approaches. For example, after erythrocyte depletion, the sample volume can be reduced by centrifugation or microfluidic techniques [57,58]. Osmotic shock (e.g., by adding distilled water) is a classical approach for lysing red blood cells. At the same time, most bacteria can survive the osmotic shock due to their rigid cell walls. Saponin and ammonium chloride are common chemical reagents for lysing red blood cells [31,59,60]. Sodium dodecyl sulfate (SDS) and Sepsityper are other examples of chemical lysis [61,62,63]. These lysis buffers and detergents are often applied in positive blood culture samples [30,64,65]. Dextran sedimentation is a physical erythrocyte depletion method, which is used for white blood cell purification [66,67]. In particular, by mixing a dextran solution with whole blood, red blood cells are depleted during sedimentation by forming rouleaux. The sedimentation process reduces the red blood cells by four orders of magnitude in 20–30 min (>20 mL/h) with over 50% capture efficiency at 10 CFU/mL [33]. An advantage of dextran sedimentation is that the process does not generate a large amount of cell debris, which can interfere with downstream processes. The removal of blood cells has also been demonstrated using physical filtration. To avoid the clogging of filters, Fang et al. designed a stirring-enhanced filtration device [29]. The filtration device removes 99.5% of the red blood cells (and all white blood cells) at ~1.2 mL/h and recovers ~70% of bacteria. The filtered samples were further concentrated by magnetic separation using microbeads coated with the flexible region of MBL and detected by PCR. 

#### 3.1.3. Acoustophoresis

Acoustophoresis refers to the migration of particles subjected to acoustic waves. In a standing acoustic wave field, a cell experiences an acoustic radiation force towards either the pressure node or the pressure antinode. The amplitude and direction of the acoustic radiation force depend on the physical properties of the cell and the surrounding medium, including size, density, and compressibility [68,69]. Most importantly, the primary acoustic force scales with the volume, i.e., the dimension to the third power, of the cell. Therefore, acoustic separation can be achieved in heterogeneous cell mixtures, such as blood, based on the size difference between the cells. Acoustic techniques have also been applied for the manipulation of various biological entities, from circulating tumor cell clusters to exosomes in the blood [32,70]. 

Toward BSI diagnostics, Ai et al. showed the separation of *E. coli* from peripheral blood mononuclear cells (PBMC) in a microfluidic sheath flow device with a purity of 95.65% [71]. The flow rate in this study was 0.03 mL/h. Ohlsson et al. reported an ACUSEP system that integrates acoustic sample preparation modules (separation and enrichment) and a dry reagent PCR microchip [35]. The system achieved a detection limit of 10^3^ CFU/mL at ~3 mL/h in blood samples spiked with *Pseudomonas putida* and successfully detected *E. coli* in half (2 out of 4) of the BSI patient samples. The entire process can be finished in less than 2 h and requires minimal manual processing. A challenge of acoustic separation is the domination of acoustic streaming for small objects (<2 µm) [72]. To mitigate the effect of acoustic streaming, Assche et al. report a gradient acoustic focusing (GAF) device that enables the separation of submicron particles and bacteria [36]. GAF is achieved by suppressing acoustic streaming using an acoustic impedance gradient with an inhomogeneous medium (Ficoll). The study reported a recovery rate of 79.77% with 10^5^ CFU/mL *S. aureus* in blood lysates (0.72 mL/h), using a combination of cell lysing and acoustic separation.

#### 3.1.4. Electrokinetics

Electrokinetics describes the motion of fluids and particles in external electric fields [73,74]. For example, dielectrophoresis (DEP) refers to the motion of polarizable objects (e.g., a cell) under a spatial electric field gradient. The dielectrophoretic force depends on the volume of the cell and the relative polarizability between the cell and the fluid. By tuning the frequency, it is possible to adjust the effective polarization and the cell motion toward (positive DEP) or away (negative DEP) from the electric field maxima [75]. External electric fields can also induce fluid motion, such as AC electrothermal flow (ACEF), in microfluidic systems [76]. ACEF creates a long-range (e.g., centimeter-scale) fluid circulation [77], which has been applied for improving the identification and AST of bacteria in blood samples [78,79].

Direct current electrokinetic techniques, such as capillary zone electrophoresis (CZE) and isoelectric focusing, can be applied for separating complex bacteria mixtures. Huge et al. reported a CZE device coupled with an automated fraction collection to separate bacteria from the salivary wastewater microbiome [80]. The bacteria were firstly separated based on their differences in electrophoretic mobility and then fractionated and cultured on agar plates for downstream analysis. This technique improves the sensitivity of bacteria detection with genome sequencing by eliminating the masking effect of the high-abundance bacteria over the low-abundance bacteria. For uncultivable bacteria, Jiang et al. introduced a recycling free-flow isoelectric focusing (RFFIEF) method-based electrophoresis method to separate the salivary microbiome [81]. After RFFIEF separation, the results showed that the commonly identified genera were retained, the low-abundance bacteria (e.g., Serratia) that cannot be detected by the conventional method were dramatically enriched, and the number of bacterial genera identified was increased by 225% on average. However, this technique can potentially be modified for improving the detection of bloodborne pathogens.

Kuczenski et al. reported a negative DEP device with electrodes tilted at shallow and steep angles along the flow direction for sorting *E. coli* from blood cells [37]. The device recovered 30% of viable cells at a flow rate of 0.035 mL/h [37]. Using interdigitated electrodes, Bisceglia et al. presented a positive DEP device with a 97% capture efficiency for *E. coli* spiked in diluted blood [38]. The device was also capable of simultaneously separating *E. coli*, *Staphylococcus epidermidis*, and *Candida albicans*. DEP is a short-range force field, which is strongest near the electrode edges [74]. By using a 3-parallel electrode design, Gao et al. demonstrated a pathogen concentration device that enhances the trapping efficiency of the DEP with long-range ACEF in the conductive fluids [39]. The device concentrates *E. coli*, *Bacillus globigii*, and *A. baumannii* in urine and buffy coats for 2–3 orders of magnitude, at a flow rate between 0.006 and 0.06 mL/h. Overall, electrokinetics has a relatively low throughput; nevertheless, DEP has a high specificity, as the dielectrophoretic force depends strongly on both the size and polarizability of the cells. This specificity is important when the purity of the sample is essential for downstream procedures in the workflow.

#### 3.1.5. Inertial Focusing

Inertial microfluidics concerns the lateral motion of particles or cells in a microchannel due to passive, inertial lift forces, which push cells away from the channel wall. The inertial lift forces are associated with fluid shear, flow disturbance near particles, and shear gradient [82,83,84]. In addition, the channel curvature and the rheological properties of the media can also create additional forces (e.g., Dean drag force and elastic force) on the cells [85]. These forces depend on the properties of cells and media and can be tuned by the microchannel design and flow rate. Since the equilibrium positions from the channel will depend on the cell types (e.g., bacteria and blood cells), inertial forces focus cells into different streamlines for separation. While typical microfluidic systems operate at a low flow rate, inertial focusing occurs at a relatively high flow rate (Reynolds number typically from 1–100), improving its significance in high throughput cell separation. 

The high-precision inertial focusing and self-ordering of red blood cells have been demonstrated in straight and curved microchannels [83]. A cross-channel design was demonstrated for removing 80% of *E. coli* (10^8^ CFU/mL) spiked in whole blood at a flow rate of 6 mL/h (240 mL/h in a 40-channel device) [40]. A spiral microfluidic device based on Dean flow fractionation reported a recovery rate of >65% for *E. coli* (10^2^ cells/mL) at ~3 mL/h. The device also separated four different bacteria of various sizes and shapes (*E. coli*, *S. aureus*, *P. aeruginosa*, *Enterococcus faecalis*) at clinically relevant concentrations (~10–50 cells/mL) [41]. Furthermore, electro-inertial fluidics can enhance the separation resolution by introducing elastic force using non-Newtonian fluids [42,86]. To sort small bacteria of similar sizes (0.5 µm to 3 µm), Lu and coworkers designed an elasto-inertial microfluidic device with periodic contractions along the spiral channel and non-Newtonian fluid. The elasto-inertial microfluidic device achieved a recovery rate of 80% for *K. pneumoniae* and 60% for *Streptococcus pneumoniae* from diluted blood at 0.3 mL/h, with a bacterial load as low as 10^2^ CFU/mL. 

Another related phenomenon is margination, which describes the accumulation of red blood cells in the center of blood vessels and the migration of white blood cells and platelets in the near-wall region [87]. The segregation of RBCs to the low shear region and other cells to the high shear region is contributed by the high deformability of red blood cells [88]. Hou and coworkers employed the principle to design a pathogen removal microfluidic device and achieved 80% and 90% removal efficiencies for *E. coli* and *Saccharomyces cerevisiae* spiked in whole blood, respectively [45]. The device demonstrates a throughput of 1 mL/h and can be multiplexed for label-free isolation.

Overall, there are some trade-offs between the throughput, recovery rate, and purity of the sample preparation techniques. For instance, inertial microfluidics has a relatively high throughput, while electrokinetics has a high selectivity between cell types. Due to the large volume mismatch between the pathogens (femtolitre) and the blood samples (milliliter), multiple methods may be combined to achieve the required reliability in clinical diagnostics. The sample preparation modules should also be combined with microfluidic detection and characterization techniques for comprehensive BSI diagnostics. Notably, the majority of reports are based on processed or spiked blood samples. The clinical applicability and reliability of these separation technologies in direct BSI diagnostics remain to be investigated. Since these methods often depend on the properties of cells and the media, the influences of sample heterogeneity and pathogen diversity should be considered. 

### 3.2. Sensitive and Quantitative Pathogen Detection and ID with Timely AST 

Other key elements of a BSI diagnosis are the ability to provide sensitive, quantitative, and accurate detections and species-level identifications of pathogens, while also offering timely and universal antibiotic susceptibility profiling in the form of an MIC.

A sufficient volume of blood per culture (typically 40 to 60 mL) is usually needed to achieve adequate and sensitive detection of as low as 1 CFU/mL of whole blood within a 24 h timeframe [89,90]. Yet, even when ample sample volume is available, the likelihood of a false negative result is not entirely negated when using blood culture due to factors such as (1) prior antibiotic usage, (2) fastidious, slow-growing, or obligate intracellular organisms, (3) the presence of pathogens other than bacteria or yeasts, and 4) culture media bias towards the growth of certain organisms [91]. Unfortunately, ID technologies such as multiplex real-time PCR (FilmArray), fluorescence in situ hybridization with peptide nucleic acid probes (QuickFISH), and matrix-assisted laser desorption ionization time of flight mass spectrometry (MALDI-TOF/MS, VITEK MS), which are available for clinical use and have less than 60 min turnaround time, still rely on positive blood culture as their sample input [92]. This requirement limits them from supplanting blood culture. 

On the other hand, the goal of AST is to guide the selection and dosage of antibiotics for effectively treating the infection. This objective can be achieved by either (1) obtaining minimum inhibitory concentration (MIC) information to predict the success of antibiotic treatment or (2) obtaining resistance information to guide antibiotic decisions. Culture-based AST alone does not always offer reliable susceptibility information. For instance, in cases where microbial biofilm is formed, a comprehensive analysis including the phenotypic measurement of biofilm production may facilitate more appropriate antibiotic choices [93,94,95].

The following highlights some of the recent advances in BSI diagnosis technologies that seek to address current gaps in ID/AST.

#### 3.2.1. Digital PCR (dPCR)

PCR has been the mainstay of many disease detection methods based on the targeted amplification of nucleic acid fragments. Since its invention and application in clinical settings, a wide variety of PCR techniques have been developed. Real-time PCR or quantitative PCR (qPCR) is the most common technique used today due to its relative speed and convenience. Unlike conventional PCR, qPCR allows the real-time quantification of nucleic acid concentration via the use of fluorescent probes. Despite its wide use in various applications, qPCR suffers from inaccuracy when it comes to detecting samples with low concentrations, and is affected by PCR inhibitors typically found in the blood. 

Digital PCR has been developed to overcome the shortcomings of qPCR by allowing the absolute quantification of nucleic acid [96]. By partitioning large samples of nucleic acid into individual reactions, dPCR pushes the lower limit of detection to a single-molecule level. Each partitioned reaction is amplified and analyzed to generate an absolute count of the target nucleic acid at the reaction endpoint, without the need for standards or internal controls. With an enhanced effective concentration in minute partition volume and the dilution of background human DNA or PCR inhibitors through sample partitioning, dPCR can greatly enhance detection sensitivity, particularly for BSI with a complex sample matrix [96].

Droplet digital PCR (ddPCR) is a version of dPCR where samples are partitioned into individual droplets [97]. Its application was the basis of a system being developed by Velox Biosystems, called the integrated comprehensive droplet digital detection (IC3D) (Figure 3C) [98]. Through a droplet microencapsulation technology, IC3D combines the capability of dPCR with DNAzyme-based sensors and a high-throughput 3D particle counter system to detect antibiotic-resistant genes from a whole blood sample. In short, unprocessed whole blood is mixed with dPCR reagents inside a microfluidic device that generates picoliter-sized droplets. Droplets containing the target bacterium become fluorescent after digital PCR is performed and the signal is detected and quantified using a high-throughput 3D particle counter system, allowing for a low limit of detection of 10 CFU/mL instead of the 1000 CFU/mL limit of qPCR [99]. 

The IC3D system also has the potential to detect antibiotic-resistant bacteria from a blood sample in less than one hour through a specific primer design [98]. The use of inhibitor-resistant Taq polymerase mutants and PCR enhancers allows for the minimal pre-analytical processing of whole blood samples. Resistant genes found in Gram-positive (*vanA*, *nuc*, and *mecA*) and Gram-negative bacteria (extended-spectrum beta-lactamase genes; *bla*_CTX-M-1_ and *bla*_CTX-M-2_, and carbapenemase genes; *bla*_OXA-48_ and *bla*_KPC_), as well as bacterial identification (*E. coli* and *Klebsiella* species) and broad bacterial detection, were tested in their proof-of-concept study [100]. Overall, the IC3D system presents itself as a potential solution for obtaining rapid pathogen ID and antibiotic resistance information.

#### 3.2.2. Universal High-Resolution Melt (U-HRM) with Pheno-Molecular AST

High-resolution melting (HRM) analysis is a simple, yet powerful and novel solution for sequence variant scanning, genotyping, and sequence matching, which can be seamlessly integrated with PCR without any post-amplification processing steps. With the use of saturating fluorescent dye, precise reaction temperature control, and software algorithms, HRM can create sequence-dependent melting curves with single-nucleotide resolution for less than one minute [101,102]. Melting temperature and curve shape depend on sequence, % GC content, length, melt domains, and sequence complementarity. The flexibility of the technique, as well as the speed, homogenous assay format, high-throughput, and low cost, allow potential wide adoption in various research and clinical disciplines. 

Over the past decade, the clinical feasibility of coupling HRM with broad-range PCR for the diagnosis of various infectious diseases has been demonstrated [103,104,105,106,107,108,109,110,111,112]. Rich melting curve profiles that enhance the breadth and analytical specificity of HRM for species-level ID were achieved when used with amplicons generated from targeting the bacterial internal transcribed spacer region (ITS). 

A machine learning curve classification algorithm allowed for the automated differentiation of bacterial species based on their unique melting curve profiles when analyzed against an archived melting curve database. Coupled with this algorithm, a 90% specificity for pathogen ID was achieved from positive blood culture. Since the reference database can be incrementally updated, the assay coverage is easily expanded to include more pathogen strains [113]. 

To date, researchers have developed “single-organism” phenotypic, growth-based AST assays based on qPCR, digital PCR, and microfluidic digital LAMP [114,115,116]. Notably, Schoepp et al. were able to shorten the antibiotic incubation time in *E. coli* to 15 to 30 min by leveraging the digital-level DNA quantification [115,116]. This pheno-molecular AST was combined using qPCR with U-HRM to enable a broad bacteria ID [117,118]. In a pheno-molecular AST, bacteria are first briefly incubated with antibiotics. The amounts of bacterial DNA, as surrogates of bacterial growths between antibiotic-treated samples and controls, are quantitatively detected and compared to reveal antibiotic susceptibilities [113].

A complete molecular workflow for a sequential ID-AST can be conducted directly from whole blood for BSI [117]. Sample preparation included steps to preferentially lyse all human red and white blood cells with centrifugation to reduce the background while enriching for target cells. The turnaround time for this complete ID/AST workflow is about 8 h, which includes a 6-h pre-enrichment, 1-h rapid ID (ITS rDNA qPCR with U-HRM), and ≥1 h rapid AST (16S rRNA RT-qPCR) (Figure 3A). A limit of detection of 1 CFU/mL was achieved for the identification of *A. baumannii*, *E. coli*, *K. pneumoniae*, and *S. aureus* by their unique melting curves. MICs of the bug–drug pairs were subsequently determined by RT-qPCR with threshold cycle, or Ct, differences of ≥1 cycle [118]. 

One of the main limitations of traditional HRM is in resolving species in a polymicrobial infection. An ensemble, composite melting curve from all the species in the mixture is impossible to decouple into individually contributing species. This limitation can be overcome by combining digital PCR with HRM to perform the absolute quantification of target cells/DNA to resolve heterogeneous populations [110,119]. 

To resolve polymicrobial infections in a spiked-in polymicrobial urine specimen, U-HRM and pheno-molecular AST can be combined on a dPCR platform to yield accurate identification of multiple bacterial species and their susceptibility profiles within ∼4 h [112]. Despite demonstrated feasibility for measuring susceptibility based on bacterial proliferation, a prolonged doubling time of fastidious species makes this approach less than ideal. To further accelerate pheno-molecular AST independent of cell division, Yang et al. discovered RNA markers that confer ciprofloxacin susceptibility as early as 10 min after antibiotic exposure and demonstrated marker expression profiles with concordant MIC results from traditional culture-based AST for multiple isolates of *K. pneumoniae* [120].

Analysis at the single-cell level offers valuable clinical information for guiding BSI therapeutic management by (1) resolving polymicrobial infections, (2) differentiating contaminants from true pathogens, and (3) correlating pathogen quantity with disease severity and treatment efficacy.

#### 3.2.3. Gamma Peptide Nucleic Acid (γPNA)

PNAs are polyamide-based synthetic nucleic acids that are capable of binding to complementary oligonucleotides with high specificity and thermal stability [121]. However, due to its non-ionic backbone, PNAs are only moderately soluble in water. Installation of the chiral center at the gamma-backbone renders the molecule more water-soluble [122]. The performance of PNA can be further enhanced by using a double-stranded configuration [123]. 

Current commercial systems, including AdvanDx QuickFISH by OpGen and AcceleratePheno, have demonstrated the capacity to deliver fast ID and AST from positive blood culture [124,125]. Leveraging on the properties of γPNA analogs with higher kinetics, sensitivity, and specificity over standard PNAs, startup company HelixBind designed a BSI diagnosis workflow that is compatible with analyzing whole blood samples [126]. Only perfect hybridization with target DNA will result in chemiluminescence that leads to optical detection (Figure 3B). Their technology incorporates a selective lysis process to almost entirely remove somatic cells without killing the microbes [126]. Comprehensive detection of more than 20 of the most common pathogens at species-level, with >95% specificity was achieved. Their entire workflow can be completed in under 2.5 h. This rapid turnaround could significantly affect the initiation of targeted BSI treatment.

#### 3.2.4. Next-Generation Sequencing (NGS)

NGS methodologies offer an all-in-one approach to identify a broad range of BSI pathogens, as well as screening for known resistance markers in suspected BSI samples. Over the past decade, several studies have employed the NGS of cell-free DNA (cfDNA) from plasma to diagnose BSI in suspected septic patients, and shown that NGS has a 93.7% agreement with and higher sensitivity than traditional culturing methods [127,128]. Apart from these advantages of NGS over blood culture, with the advent of real-time Nanopore sequencing, pathogens can be identified within minutes of sequencing and the entire workflow can be achieved within six hours of blood draw [129]. 

The utility of NGS expands beyond simply identifying the BSI. Metagenomic sequencing on cfDNA facilitates screening for resistance markers as well [129]. However, identifying resistance genes does not correlate with actual resistance. Conferred resistance phenotype depends on the activation of these genes and does not suggest a MIC for the preferred antibiotic [130]. Known resistance mechanisms are limited and will continue to evolve. A secondary approach would be predicting MIC from assembled whole genomes using predictive models. Such models have been described for *Neisseria gonorrhoeae* and *Salmonella enterica* [131,132]. Comparing whole genomes of *N. gonorrhea* allows for the determination of single nucleotide polymorphisms (SNPs) in known molecular antimicrobial resistance determinants and the discovery of novel susceptibility loci using genome-wide association studies (GWAS) [131,133]. The observed range of MICs of *N. gonorrhoeae* to cephalosporins could be attributed to non-synonymous substitutions in *penA*, *porB*, *ponA*, and a disrupted mtrR promoter and the resulting model predicted MICs with an overall sensitivity and specificity of 99.9% and 97.1%, respectively [134]. Similarly, whole genomic sequencing-based AST was shown to have an 89.8% concordance between predicted and experimental MIC in *S. enterica* for an array of 5 antimicrobials with a sensitivity and specificity of 89% and 97%, respectively [132]. Aside from a pathogenic-centric approach, NGS technologies like assay for transposase-accessible chromatin (ATAC)-seq on specific human cell types can provide sensitive pathogen detection with concurrent host response signal to the infection because only accessible host chromatin regions are sequenced. ATAC-seq on human neutrophils challenged with *S. aureus* was shown to have greater sensitivity in detecting bacterial reads (10^3^ CFU/mL) in comparison to the traditional library preparation methods (10^5^ CFU/mL) while simultaneously detecting host epigenomic responses to the pathogen [135]. 

The potential utility of NGS in ID/AST of BSI is undeniable but the application of the technology for this purpose is in its infancy. Despite the many advantages of NGS over blood culture like the breadth of information obtained from a single run or the speed of accruing this information, it has its caveats. Limitations of this technology are its dependence on extensive and well-curated databases to interpret the generated data and the choice of thresholds adopted; the inability to differentiate between DNAemia and bacteremia; and the clinical interpretation of whether the detected organisms are indeed infectious or are commensals, colonizers, or just contamination. To accurately identify the pathogen causing the BSI, reads generated on an NGS platform need to be aligned against a reference database that is comprehensive and can delineate between closely related species and potentially between strains within a species. For example, the clinically validated Karius test utilizes an advanced machine learning algorithm to analyze genomic data from cfDNA against their proprietary, constantly refined, a reference database of more than 1000 clinically relevant species of bacteria, fungi, parasites, and viruses (Figure 3D) [128]. Similar databases are required for resistance markers and whole-genome sequences of various pathogenic species to be able to predict the MIC accurately. While these databases are currently limited, the addition of more such data and the creation of analysis pipelines that combine both ID and AST will only strengthen the utility of NGS in BSI diagnosis in the near future. 

#### 3.2.5. Surface-Enhanced Raman Spectroscopy

Raman spectroscopy is a non-invasive, label-free, real-time analysis tool based on the vibrational properties of materials [136]. The spectrum obtained from the spectroscopy measurement serves as a molecular fingerprint for the analyte. The major hurdle in the use of Raman spectroscopy is in its ability to achieve highly accurate results in complex samples [137]. The relatively low probability and weak Raman scattering efficiency from bacterial cells could easily be masked by background noise, hence ruling out its potential as a sensitive diagnosis method. In addition, a long measurement time is usually required to increase the signal-to-noise ratio, making it unsuitable for high throughput analysis [137]. 

To overcome this hurdle, Raman signals have been improved through the use of metallic surface enhancement via surface plasmon resonance. In recent decades, metallic nanoparticles have improved the performance of Raman spectroscopy, allowing measurement with higher signal intensity, resolution, and limits of detection at a single-molecule level. Rapid pathogen detection in blood using surface-enhanced Raman spectroscopy (SERS) has received increased interest for AST measurement by observing spectral changes corresponding to the antibiotic metabolism [136,138].

Recent developments in advanced statistical data analysis and sample preparation techniques have helped to redeem the utility of SERS-based pathogen detection and ID. Utilizing electrokinetic methods, Cheng et al. were able to efficiently separate and concentrate bacteria cells from diluted blood by applying a fine-tuned AC voltage on a set of electrodes. They reported a rapid pathogen identification in less than 5 min using this SERS (Cheng et al., 2013). By using a convolutional neural network, Ho and coworkers were able to generate accurate identification of 30 common pathogens with identification accuracies of up to 99.7% from Raman spectra [139,140]. 

A commercial system by spectral platform (Spectral-01) uses Raman spectroscopy as its main technology for all-in-one ID and AST. The company claims to detect the pathogen in samples with as low as <1 CFU/mL pathogen load (according to their press release). Their detection method relies on the interaction between bacteria (positively charged cell wall) and human serum albumin (negatively charged surface). Free radicals produced by the bacteria metabolism oxidize the cysteine group of the albumin molecules into albumin dimer. The release of free radicals is reflected by the decrease in Raman and lycopene fluorescence signals, confirming the presence of bacteria.

The assay setup is intended for translation into ID and AST application. So far, a pathogen ID specificity of 94% for over 30 pathogens was obtained in under 20 min (according to their press release). The presence of specific bacteria can be identified by incorporating specific disulfide crosslinkers that can only be cleaved by enzymes produced by certain bacteria. Bacterial susceptibility to antibiotics is then measured by monitoring the level of Raman and fluorescence signals. 

The clinical utility of SERS-based ID and AST as a universal diagnosis approach will rely on the availability of a robust spectral database. Quantification of pathogen load may be possible by measuring changes in specific net signals in the sample. While current developments are still in the early stages, the single-cell sensitivity and fast turnaround of Raman-based measurement show promise as future culture-free BSI diagnosis approach for meeting critical clinical timepoint [138,141]. Further development in this direction will determine their translational feasibility in clinical settings.

#### 3.2.6. Flow Cytometry

Flow cytometry is a cellular analysis method widely used for studying cellular characteristics [142]. Individual cells in a liquid suspension flow through a fluidic system and pass a laser light source using laminar flow. Scattered lights are detected by photomultiplier tubes. Lights scattered at an acute angle (forward scatter), due to light diffraction upon contact with the cell surface, give information about the cell size, while wide (90°) angle light scattering (side scatter) from refracted light at the interface between the laser and the intracellular structure is indicative of the cells’ roughness and granularity. Using this technique, information such as cell sizes, physiological conditions, and protein content can be obtained. 

While the use of flow cytometry in microbiology is growing, its clinical application is still limited. A startup founded in 2013, FASTinov, recently patented a flow cytometry-based antibiotic susceptibility technology (FAST) that identifies and differentiates carbapenemase production in *Enterobacteriaceae*, by measuring the fluorescence intensity of a fluorochrome dye [143]. The susceptibility information correlated (98%) with the standard AST method [144]. The company also demonstrated the use of this platform to determine the MIC of bacteria challenged with colistin after only 1 h of incubation. In short, a 96-multiwell AST panel consisting of dehydrated colistin at a serial concentration of 0.125 to 65 μg/mL and a fluorescent dye is inoculated with bacterial suspension. After 1 h of incubation, bacterial cell fluorescence intensity and morphologies were analyzed using a flow cytometer. The in-house software translates the cytometer readouts into MIC values by incorporating the number of events, light scattering patterns, and fluorescence intensity of each well onto predetermined cut-off values. In a study involving 116 Gram-negative bacilli, the authors demonstrated a highly reproducible (97%), automatically generated MIC result after 1.5 h [143]. 

The fast <2 h turnaround, with MIC information from positive culture (equivalent to an overall <26 h turnaround) puts the FASTinov platform ahead of conventional AST methods that take at least 30 h to complete. However, the reliance on blood culture and the lack of ID capacity would set the system apart from other emerging platforms.

## 4. Summary and Future Outlook (Conclusions)

An accurate and timely diagnosis of BSI is important for promoting the appropriate use of antibiotics. The lack of rapid ID/AST in current clinical approaches warranted the initiation of antibiotic treatment without a confirmed diagnosis. To date, the T2 Biosystem is the first and only FDA-approved system capable of identifying a BSI pathogen directly from whole blood. So far, no commercial systems could perform both ID and AST for diagnosing BSI directly from whole blood samples using low sample volume and in a rapid manner. Nevertheless, promising new technologies are emerging and paving the way for efficient, sensitive, selective, and comprehensive diagnoses. 

Our assessment of the current ID/AST developmental progress (Figure 4, Table 3) reveals the potential for several emerging techniques to achieve highly sensitive and specific ID/AST for BSI. From our current assessment, qPCR-HRM, SERS, IC3D, and γPNA-FISH appear to be the most promising in terms of delivering highly sensitive and specific ID/AST from whole blood samples. Their relatively fast turnaround likely fits into the critical clinical timepoints concerning antibiotics selection, escalation, and de-escalation. However, only the qPCR-HRM workflow was able to deliver MIC information, and only γPNA-FISH could deliver results from whole blood in less than three hours. These features are particularly desirable for guiding clinical decisions.

While novel approaches with rapid outcomes are a priority, discrepant results derived from new methods against imperfect conventional methods must be interpreted with caution. The clinical significance of microbial sequences identified in the blood must be interpreted along with the clinical context. For accelerated AST, the growth kinetics of the bacteria in the presence of antibiotics could be confounded by variations in inoculum size from isolated colonies, the presence of dead cells, and the growth phase of the inoculated bacteria [145]. Furthermore, isogenic bacterial populations, shown to have heteroresistance, are prone to the false categorization of susceptibility [146,147], and in some isolates, antibiotics exposure can easily enhance antibiotic resistance [148]. Therefore, understanding factors affecting antibiotic responses such as potential inoculum effect, or delayed resistance due to variation in growth phases or delayed induced resistance expression, should be taken into account when designing a new assay that relies on a vastly different format from the current reference standard [12].

Although we have only discussed microbiological technologies, combination analysis that takes into account host responses (including immunity and hemodynamics) could be a strategy for improving future BSI diagnosis as well as the prognosis of clinical outcomes. Future development towards fulfilling the four key elements of an ideal BSI diagnosis should consider integrating suitable sample preparation techniques to enhance downstream analysis. We anticipate that future breakthroughs will likely arise from technologies that require (1) minimal handling and analysis of the low pathogen, (2) the separation, enrichment, and concentration of pathogens or target analytes from background interferences, (3) the delivery of sensitive, quantitative, and agnostic detection and broad species-level identification and (4) the provision of timely universal antibiotic susceptibility profiling with MIC reporting that matches critical clinical decision timepoints. 

All in all, it remains unclear if a “one-size-fits-all” test with superior performance characteristics will ever supplant blood culture practice in BSI diagnosis. While exciting development is underway, it is easier to envision these technologies augmenting the reduction in early diagnostic uncertainty to impact clinical outcomes. 

## Figures and Tables

**Figure 1 antibiotics-11-00511-f001:**
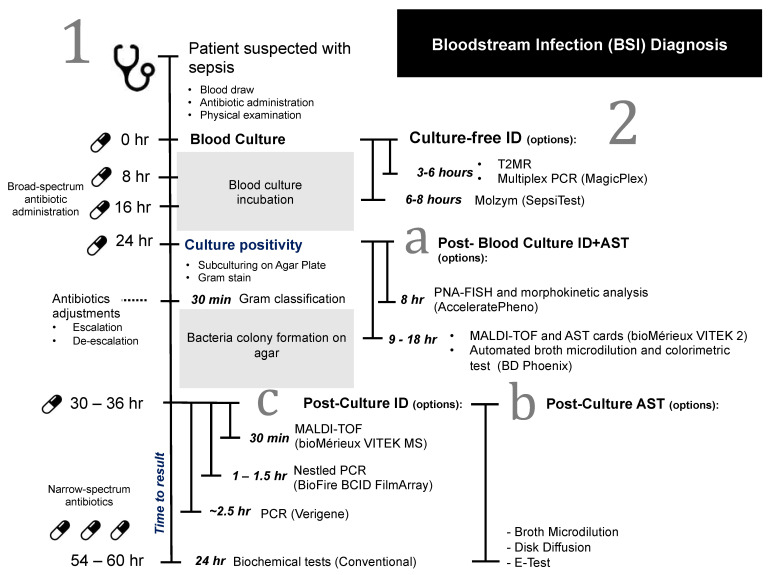
Workflow of current BSI diagnosis. (1) Blood collected from patients is subject to culturing followed by either an all-in-one post-culture ID/AST (a) or post-culture ID with a separate AST (b + c). (2) Culture-free ID workflow is also currently available with a separate AST from positive culture. Antibiotics administration at the point of examination. Antibiotic choices may be adjusted accordingly.

**Figure 2 antibiotics-11-00511-f002:**
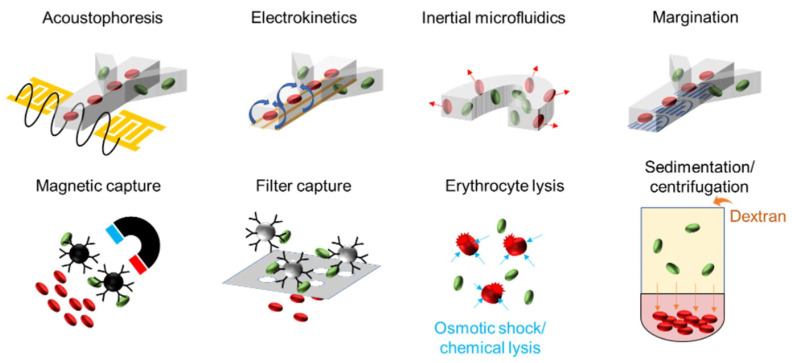
Sample preparation techniques for BSI diagnostics. Pathogen isolation and concentration can be achieved actively via acoustophoresis, electrokinetics, and magnetic forces, or passively via inertial focusing, margination, filtering, and erythrocyte lysis.

**Figure 3 antibiotics-11-00511-f003:**
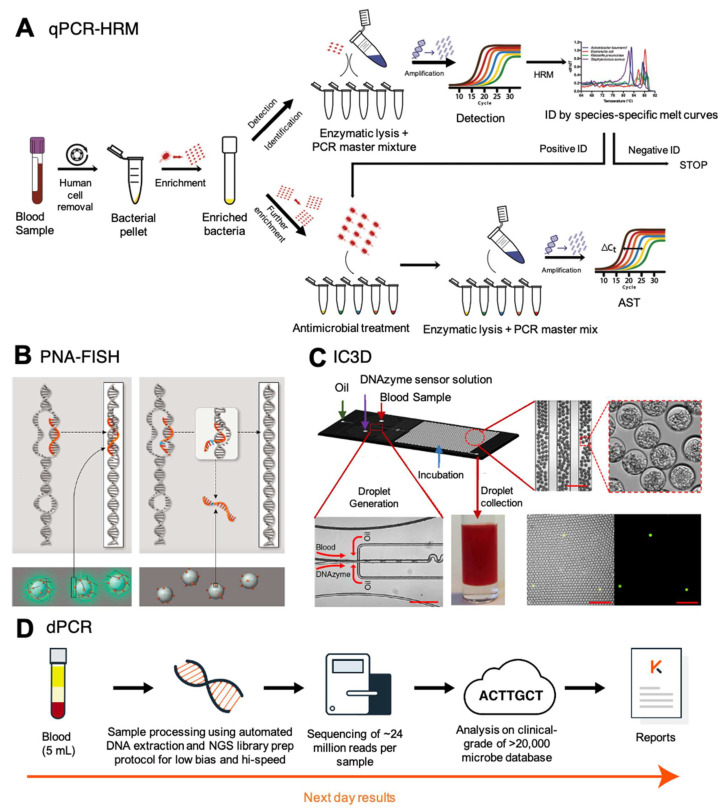
Recent advancements to improve BSI diagnosis. (**A**) qPCR-HRM workflow from a whole blood sample. (**B**) HelixBind PNA-FISH technology with a sequence-specific fluorescence dye, (**C**) Microfluidic-assisted whole blood compartmentalization before DNA amplification and analysis, a trademark of IC3D technology (**D**) NGS detection workflow directly from whole blood by Karius. Figures were modified and used with permission from (Andini et al., 2018; Blauwkamp et al., 2019; D. K. Kang et al., 2014; Nölling et al., 2016).

**Figure 4 antibiotics-11-00511-f004:**
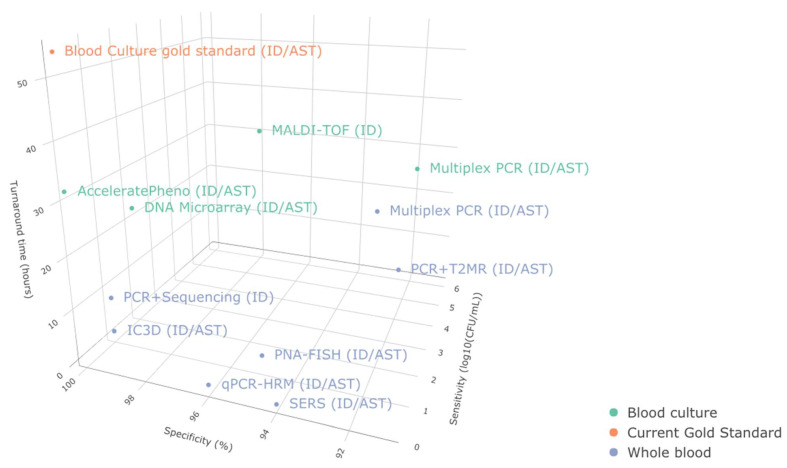
Performance characteristics of current and emerging technologies. Sensitivity, specificity, and turnaround time compared against the current gold standard. 1. qPCR-HRM, 2. SERS, 3. IC3D, 4. PNA-FISH 5. PCR+T2MR, 6. Multiplex PCR from whole blood, 7. PCR + sequencing, 8. Multiplex PCR from blood culture, 9. DNA Microarray, 10. MALDI-TOF, 11. AcceleratePheno, and 12. Blood culture gold standard.

**Table 1 antibiotics-11-00511-t001:** List of commercial and developing technologies for BSI diagnosis.

No.	Company	System	Approach	Status *	Sample Prep ^‡^	Detection/ ID	AST	TAT ^^^
1	Abacus Diagnostica	Genomera CDX	Rapid/ Real-Time PCR	Dev.	BC (+)	✓		50 min
2	Affinity Biosensors	LifeScale AST	Microorganism mass measurement	CE-IVD	BC (+)		✓	4 h
3	Amplex Diagnostics, GmbH, Germany	Eazyplex MRSA	LAMP ultra-rapid MRSA detection	CE-IVD	BC (+)	✓		30 min
4	Arc Bio	Galileo pathogen solution	Shotgun Sequencing	Dev.	WB	✓		48 h
5	BD	GeneOhm MRSA	Real-Time PCR	FDA, CE-IVD	BC (+)	✓		2 h
6	Becton Dickinson	BD Max StaphSR	Real-Time PCR	FDA	BC (+)	✓		~1.5 h
7	BioFire/bioMerieux Diagnostics	FilmArray DIRECT (new)	Nested PCR	FDA, CE-IVD	WB	✓		1 h
8	BioRad	Droplet dPCR	dPCR; absolute quantification using Poisson’s statistics without requiring a standard curve	CE-IVD	BC (+)	✓		No report
9	BioSense Solutions (Denmark)	oCelloScope	3D optical scanning microscopy imaging	Dev.	BC (+)		✓	1 to 4 h
10	Bruker Daltonics	MALDI Biotyper + DxM MicroScan WalkAway System	Mass spectrometry	FDA, CE-IVD	BC (+)	✓		12 to 24 h
11	DNAe (electronic)	LiDia Bloodstream Infection Test	WGS/NGS/miniaturised sequencing	Dev.	WB	✓		3 to 4 h
12	FASTinov	Flow cytometry	Cell sorting fluorescence-based AST	Dev.	BC (+)		✓	<2 h
14	Roche	Smarticles	Bacteriophage-based	Dev.	BC (+)	✓		No report
15	GenMarkDx USA	ePlex BCID	Multiplex PCR	CE-IVD	BC (+)	✓		1.5 h
16	Gradientech AB	Rapid IVD; QuickMIC and CellDirector	Microfluidics Phenotypic multiplex chip	Dev.	BC (+)		✓	2 h
17	Great Basin Corporation (Bringham Young Univ.)	OptoFluidic Platform	Single molecule fluorescence hybridization	Dev.	WB	✓		1 h
18	Hologic	AccuProbe	In situ hybridization	CE-IVD	BC (+)	✓		1 h
19	iCubate	iC GPC	Multiplex amplification assay	FDA, CE-IVD	BC (+)	✓		4 to 5 h
20	IRIDICA	BAC BSI Assay	PCR/ESI-MS	withdrawn	WB	✓		8 h
21	Karius, Inc.	Karius Test	NextGen Seq cfDNA; Genomic; Bioinformatics	Dev.	WB	✓		48 h
22	Luminex	Verigene Gram+ BC	Microarray	FDA.	BC (+)	✓		2.5 h
23	Luminex	Verigene Gram− BC	Microarray	FDA	BC (+)	✓		2.5 h
24	Master Diagnostica, Spain	Sepsis Flow Chip	Microarray	CE-IVD	BC (+)	✓		3 to 4 h
25	Molzym, Germany	SeptiTest; UMD SelectNA	Real Time PCR	CE-IVD	WB	✓		8 to 12 h
26	Momentum Biosciences (Cardiff, UK)	TBD Cognitor Minus	Enzymatic template generation and amplification	awaiting clearance	BC (+)	✓		No report
27	OpGen USA	PNA FISH	In situ hybridization	CE-IVD	BC (+)	✓		2.5 h
28	OpGen USA	Quick FISH	In situ hybridization	CE-IVD	BC (+)	✓		30 min
29	QLinea (Uppsala, Sweden)	AsTAR	High-speed time-lapse microscopy imaging of bacteria in broth	Dev.	BC (+)		✓	6 h
30	Resistell (Switzerland)	Rapid AST antibiogram	AFM, Cantilever, Nanomotion detection-based AST	unknown	BC (+)		✓	No report
31	Roche Molecular System, Switzerland	LightCycler SeptiFast	Real-Time PCR	CE-IVD	WB	✓		6 h
32	SeeGene, Korea	Magicplex Sepsis RT test	Real-Time PCR	CE-IVD	WB	✓		3 to 6 h
33	Specific Diagnostics Inc	Reveal phenotypic AST	Detection of volatile organic compounds	Dev.	BC (+)		✓	~5 h (with MIC)
34	T2Biosystem	T2 Candida Panel T2MR	Nuclear Magnetic Resonance	FDA, CE-IVD	WB	✓		3 to 5 h
35	QuantaMatrix	QMAC-dRAST	Optical Microscopy	Dev.	BC (+)		✓	4 to 6 h

* Platforms on this list are either U.S. Food and Drug Administration (FDA) and/or European CE Marking for In Vitro Diagnostic (CE-IVD) certified or under research development (Dev.); ^‡^ BC (+): blood culture-positive; WB: whole blood; ^^^ TAT: turnaround time.

**Table 2 antibiotics-11-00511-t002:** Comparison of microfluidic techniques for isolating bloodborne pathogens. Values are estimated for a single channel with a single pass.

Isolation Method	Recovery Rate ^#^	Throughput ^##^	Pathogens	Sample Type	Conc. (cell/mL)	Ref
Affinity capture (magnetic; antibody)	78%	0.025 mL/h	*E. coli*	Red blood cells	5 × 10^6^	[26]
Affinity capture (magnetic; antibody)	80%	20 mL/h	*C. albicans*	Whole blood	1 × 10^6^	[27]
Affinity capture (magnetic; lectin)	60–90%	10 mL/h	*S. aureus, C. albicans, E. coli*	Whole blood	1 × 10^4^	[28]
Filtration Affinity capture (magnetic; MBL)	68–76%	1.2 mL/h	*E. coli, P. aeruginosa, K. pneumoniae, S. saprophyticus, S. epidermidis*	Whole blood	10^1^–10^2^	[29]
	56–77%	3 mL/h		Filtered blood		
Affinity capture (magnetic; Zn-DPA)	>88%	60 mL/h	*E. coli*	Whole blood	5 × 10^6^	[30]
Erythrocyte depletion (detergent + water)	~100%	2.88 mL/h	*E. coli, M. luteus*	Whole blood	1 × 10^7^	[31]
Erythrocyte depletion (lysis)	>90%	20 mL/h	*E. coli*	Whole blood	1 × 10^3^	[32]
Erythrocyte depletion (dextran sedimentation)	50–60%	20 mL/h	*E. coli, E. faecalis, K. pneumoniae*	Whole blood	10^1^–10^2^	[33]
Acoustophoresis	95.65%	0.03 mL/h	*E. coli*	PBMC	3 × 10^6^	[34]
Acoustophoresis	91%	3 mL/h	*P. putida, E. coli*	Diluted blood	5 × 10^5^	[35]
Acoustophoresis (GAF)	79.77%	0.72 mL/h	*S. aureus, S. pneumoniae, E. coli*	Blood lysates	1 × 10^5^	[36]
Electrokinetics (DEP)	30%	0.035 mL/h	*E. coli*	Red blood cells	1 × 10^6^	[37]
Electrokinetics (DEP)	97%	0.0009 mL/h	*E. coli, S. epidermidis, and C. albicans*	Diluted blood	1 × 10^4^	[38]
Electrokinetics (DEP and ACEF)	30–80%	0.006–0.06 mL/h	*E. coli, A. baumannii, B. globigii*	Buffy coat	1 × 10^5^	[39]
Inertial focusing	>60%	12 mL/h	*E. coli*	Whole blood	1 × 10^8^	[40]
Inertial focusing (Dean flow)	>65%	0.6 mL/h	*E. coli, S. aureus, P. aeruginosa, E. faecalis*	Diluted blood	1 × 10^1^	[41]
Elasto-inertial	76%	0.03 mL/h	*E. coli*	Whole blood	1 × 10^6^	[42]
Elasto-inertial	>80%	0.3–1.5 mL/h	*E. coli, S. capitis*	Diluted blood	1 × 10^3^	[43]
Elasto-inertial	60–80%	0.3 mL/h	*K. pneumoniae, S. pneumoniae*	Diluted blood	1 × 10^2^	[44]
Margination	80–90%	1 mL/h	*E. coli and S. cerevisiae*	Whole blood	1 × 10^6^	[45]

^#^ Removal rates are reported for blood cleansing devices. ^##^ Throughputs for diluted blood samples are adjusted for compassion with whole blood.

**Table 3 antibiotics-11-00511-t003:** Comparison of Existing and Emerging BSI Diagnosis Technologies.

Technologies	Sample	Company	ID				AST	
			Sens. (CFU/mL)	Spec.	Breadth	TAT	Output	TAT
*EMERGING*								
*qPCR-HRM*	WB *	Non-commercial	1	100%	37 bacteria (expandable)	8 h (with AST)	MIC	8 h (with ID)
*SERS*	WB	Spectral Platforms	1	94%	>30 pathogens	20 min	S/R (enzyme-based)	unspecified
*ddPCR/ IC3D*	WB	Velox Bio	10	100%	unspecified	1–4 h (with AMR)	resistance marker	1–4 h (with ID)
*Flow Cytometry*	BC (+)	FASTinov	N/A	N/A	N/A	N/A	MIC	<26 h
*PNA-FISH*	WB	HelixBind	<10	95%	21 pathogens	2.5 h (with AMR)	resistance marker	2.5 h (with ID)
** *EXISTING* **								
*PCR+T2MR*	WB	T2 Biosystems	1–10	91%	5 candida species, ESKAPE organisms	27–29 h (with AMR)	resistance marker	27–29 h (with ID)
*Multiplex PCR*	WB	MagicPlex (SeeGene)	30	66–92%	>90 pathogens with 27 pathogens at species level	27–30 h (with AMR)	resistance marker	27–30 h (with ID)
*Real-time PCR+Sequencing*	WB	SepsiTest (Molzym)	10–40	86–100%	>1350 pathogens	30–31 h	N/A	N/A
*Multiplex PCR*	BC (+)	BioFire (FilmArray)	10^6^ to 10^8^	82–92%	8 Gram+/11 Gram−/5 fungi	25 h (with AMR)	resistance marker	25 h (with ID)
*DNA Microarray*	BC (+)	Luminex (Verigene)	10–100	84–99%	8 Gram+/5 Gram−	26.5 h (with AMR)	resistance marker	26.5 h (with ID)
*MALDI-TOF + AST cards*	BC (+)	Biomerieux (VITEK 2)	10^6^	61–98	1316 pathogens	30–36 h (with AST)	MIC	30–36 h (with ID)
*PNA FISH +* *morphokinetic cellular analysis*	BC (+)	Accelerate Diagnostics (Accelerate Pheno)	0.8 to 1.7	86–100	7 Gram+/8 Gram−/2 fungi	32 h (with AST)	MIC	32 h (with ID)
*Traditional Blood Culture*	WB	BD (BACTEC)	1	100%	Broad	30 h	MIC	54 h (with ID)

* WB: direct from whole blood, BC (+): from positive blood culture; Sens.: sensitivity; Spec.: specificity; TAT: Turnaround time.

## Data Availability

Not applicable.

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
