# Peer review of "Diagnosis of Bloodstream Infections: An Evolution of Technologies towards Accurate and Rapid Identification and Antibiotic Susceptibility Testing"

_antibiotics, 2022, doi:10.3390/antibiotics11040511_

Round 1

Reviewer 1 Report

 The study shows interesting findings and improve the knowledge in the field

- The paper deals with the topic of actuality.

- The paper is clear, well written, and the organization is very good.

- The references are up to date, and they are well organized according to the format required by the journal.

Author Response

Response to Reviewer 1 Comments

Point 1: The study shows interesting findings and improve the knowledge in the field

Response 1: We thank the reviewer for their kind words

Point 2: The paper deals with the topic of actuality.

Response 2: We agree with the reviewer

Point 3: The paper is clear, well written, and the organization is very good.

Response 3: We appreciate the reviewer’s comment

Point 4: The references are up to date, and they are well organized according to the format required by the journal.

Response 4: We appreciate the reviewer’s comment

Reviewer 2 Report

In this review the authors focus on BSI and highlight both the progress and shortcomings of its current diagnosis. We have looked at clinical workflows using recently approved technologies and have shown that while offering greater sensitivity and selectivity, these techniques are still unable to deliver a timely result. We then discuss a number of emerging technologies that have the potential to reduce overall response time to BSI diagnosis through direct whole blood testing while maintaining, if not improving, current test sensitivity and pathogen coverage. We concluded by providing our assessment of potential future directions to accelerate BSI pathogen identification and antibiotic susceptibility testing. Although engineering solutions have enabled faster testing, further progress is still needed to supplant the practice of blood culture and guide the administration of appropriate antibiotics for patients with BSI.

1 - Introduction: the contents and the drafting of the general part must be reformed to review the syntax of the topic.

2- Discussion:

to deepen in consideration of the problem of antimicrobial resistance and biofilm in blood and systemic infections, the support of the phenotypic analysis of resistance to formulate a faster diagnosis. Find out more about this by using and citing the following references: 

PMID: 34572716 ; PMID: 35169997 ; PMID: 34406812 

3 - Check the bibliographic entries throughout the text, some of which do not conform, review some entries in the references and necessarily insert those referred to in point 2 for the purpose of my acceptance.

4 - Review English grammar and in particular applied scientific English: in particular verb tenses and syntax in the discussion.

Author Response

Point 1: In this review the authors focus on BSI and highlight both the progress and shortcomings of its current diagnosis. We have looked at clinical workflows using recently approved technologies and have shown that while offering greater sensitivity and selectivity, these techniques are still unable to deliver a timely result.We then discuss a number of emerging technologies that have the potential to reduce overall response time to BSI diagnosis through direct whole blood testing while maintaining, if not improving, current test sensitivity and pathogen coverage. We concluded by providing our assessment of potential future directions to accelerate BSI pathogen identification and antibiotic susceptibility testing. Although engineering solutions have enabled faster testing, further progress is still needed to supplant the practice of blood culture and guide the administration of appropriate antibiotics for patients with BSI.

Response 1: We thank the reviewer for summarizing the content of our review

Point 2: Introduction: the contents and the drafting of the general part must be reformed to review the syntax of the topic.

Response 2: We appreciate the reviewer’s comments and have made amendments to our manuscript to include proper syntaxes. The corrections can be found in the revised version of our manuscript.

Point 3: Discussion:

to deepen in consideration of the problem of antimicrobial resistance and biofilm in blood and systemic infections, the support of the phenotypic analysis of resistance to formulate a faster diagnosis. Find out more about this by using and citing the following references: PMID: 34572716; PMID: 35169997; PMID: 34406812 

Response 3: We appreciate the reviewer’s input and have reviewed the suggested references (PMID: 34572716; PMID: 35169997; PMID: 34406812). We have added the following to our discussion under section 3.2:

“Culture-based AST alone does not always offer reliable susceptibility information. For instance, in cases where microbial biofilm is formed, comprehensive analysis including phenotypic measurement of biofilm production may facilitate more appropriate antibiotic choices. [92-94]”

Point 4: Check the bibliographic entries throughout the text, some of which do not conform, review some entries in the references and necessarily insert those referred to in point 2 for the purpose of my acceptance.

Response 4: We appreciate the reviewer’s comment and have checked the bibliography entries throughout the text.

Point 5: Review English grammar and in particular applied scientific English: in particular verb tenses and syntax in the discussion.

Response 5: We have reviewed and corrected grammatical errors found in our manuscript and applied changes to reflect appropriate verb tenses and syntaxes in the Discussion section.

Reviewer 3 Report

This review covered the evolution of technology for diagnosing of bloodstream infections. The current and new BSI technologies are explain and classify for their ID and AST capabilities. For each method, the ability to provide sensitive, quantitative and accurate detection and species-level identification of pathogens are essential and discussed deeply. The article is well written and easy to understand for a wide audience. Finallt, the addition of a chapter, in section 3.2, dedicated to capillary zone electrophoresis would be a further advantage for the article.

Author Response

Response to Reviewer 3 Comments

Point 1: This review covered the evolution of technology for diagnosing of bloodstream infections. The current and new BSI technologies are explain and classify for their ID and AST capabilities. For each method, the ability to provide sensitive, quantitative and accurate detection and species-level identification of pathogens are essential and discussed deeply.

Response 1: We thank the reviewer for their kind words

Point 2: The article is well written and easy to understand for a wide audience.

Response 2: We agree with the reviewer

Point 3: Finally, the addition of a chapter, in section 3.2, dedicated to capillary zone electrophoresis would be a further advantage for the article.

Response 3: We appreciate the reviewer’s comment and have added an additional discussion on capillary zone electrophoresis, which can be found under section 3.1.4 of the manuscript:

"Direct current electrokinetic techniques, such as capillary zone electrophoresis (CZE) and isoelectric focusing, can be applied for separating complex bacteria mixtures. Huge et al. reported a CZE device coupled with an automated fraction collection to separate bacteria from the salivary wastewater microbiome [82]. The bacteria were firstly be separated based on their differences in electrophoretic mobility and then fractionated and cultured on agar plates for downstream analysis. This technique improves the sensitivity of bacteria detection with genome sequencing by eliminating the masking effect of the high-abundance bacteria over the low-abundance bacteria. For uncultivable bacteria,  Jiang et al. introduced a recycling free-flow isoelectric focusing (RFFIEF) method-based electrophoresis method to separate salivary microbiome [83]. After RFFIEF separation, the results showed that the commonly identified genera were retained, the low-abundance bacteria (e.g., Serratia) that cannot be detected by the conventional method were dramatically enriched and, the number of bacterial genera identified was increased by 225% on average. However, this technique can potentially be modified for improving the detection of bloodborne pathogens."

Reviewer 4 Report

The article has an extensive review of the state of the art of diagnostic methods for detecting pathogens in blood.
It is supported by solid evidence.
Only minor spelling details are indicated in the attached file.

Author Response

Point 1: The article has an extensive review of the state of the art of diagnostic methods for detecting pathogens in blood. It is supported by solid evidence.

Response 1: We thank the reviewer for their kind words

Point 2: Only minor spelling details are indicated in the attached file.

Response 2: We appreciate the reviewer for pointing this out. We have reviewed these spelling errors and corrected them in our revised manuscript

Round 2

Reviewer 2 Report

Been made the corrections.

Author Response

Reviewer's comment: Been made the corrections.

Response: We thank the reviewer for confirming our corrections.